# The Importance of Prevention When Working with Hazardous Materials in the Case of Serpentinite and Asbestos When Cleaning Monuments for Restoration

Dolores Pereira [1],* , Ana Jesús López [2] , Alberto Ramil [2] and Andrea Bloise [3]

1   Department of Geology, University of Salamanca, 37008 Salamanca, Spain
2   Ferrol Industrial Campus, Universidade da Coruña, 15471 Ferrol, Spain
3   Department of Biology, Ecology and Earth Sciences, University of Calabria, I-87036 Rende, Italy
*   Correspondence: mdp@usal.es

**Abstract:** Health risks are often overlooked when the consequences are not evident in the short term. In restoration work, some activities can generate particles that may affect the health of workers through inhalation (e.g., cleaning of buildings or heritage artifacts composed of stone). Workers at quarries are also exposed to such materials and, therefore, the results of our work can help to increase the risk perception in workers from the stone sector, but also in construction workers in an environment associated with dust. To demonstrate the importance of protection to prevent health hazards, we laser-ablated some samples of serpentinite that contain serpentine minerals as major phase minerals. The powder obtained in filters coupled to the ablation laser was analysed, using tools such as an optical microscope, X-ray powder diffraction, a transmission electron microscope and thermal analysis. The results were very didactic, and the intention is to use them, by way of graphics and diagrams, to build information security sheets that will alert workers to the need of using masks when working, or to use tools with coupled filters such as the one used for our study. The main goal is to demonstrate that the interdisciplinary combination of scientific approaches can be used for the sake of human health.

**Keywords:** asbestos; health risks; laser ablation cleaning; heritage stones; occupational safety

## 1. Introduction

Natural stones are important raw materials, used since ancient times to build human shelter, but also to build our cultural heritage. Serpentinites are important rocks that have been used worldwide for these purposes [1]. From a mineralogical point of view, serpentine-group minerals (i.e., chrysotile, lizardite, antigorite) are the main constituents of serpentinite rocks [2], although serpentinites may also contain accessory minerals such as chlorite, brucite, magnetite, talc and/or carbonates, as well as remnants of the precursor mineralogy, such as pyroxene and olivine [2]. Serpentinites have a complex origin, involving deep parts of the Earth, including the mantle. They are formed as a result of the serpentinization of ultramafic rocks (i.e., lherzolite, harzburgite and dunite), in which the main minerals are pyroxenes and olivine. Antigorite, lizardite and chrysotile are polymorphs with a similar chemical composition, close to the magnesian end-member $Mg_3Si_2O_5(OH)_4$ [2]. There are very important outcrops of serpentinites around the world that have been the subject of extensive scientific research regarding their formation and geological interest [3,4]. However, these stones, if inappropriately used, pose a danger to human safety, as they may contain asbestos. The term "asbestos" refers to a group of fibrous silicate minerals, which include serpentine (chrysotile) and amphibole minerals (tremolite, actinolite, anthophyllite, amosite, crocidolite) [2]. There is extensive literature referring to serpentinites from Italy, where these rocks have traditionally been marketed as ornamental stones. Moreover, in India and Spain, serpentinites were widely used to

build an extraordinary heritage site, which could be potentially endangered if deterioration affects the stone (Figures 1 and 2) [3–7].

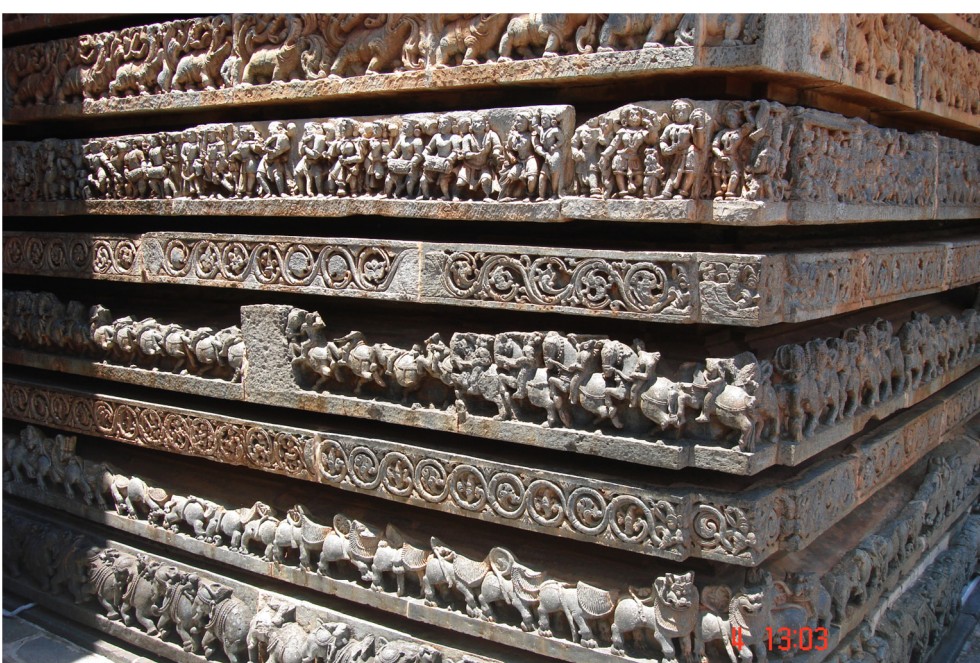

**Figure 1.** Religious monument in India showing an elaborate carving in soapstone, a much further evolution of an ultramaphic rock composed mainly of talc and serpentine minerals, which allows easy sculpting. Picture courtesy of J. Mudlappa.

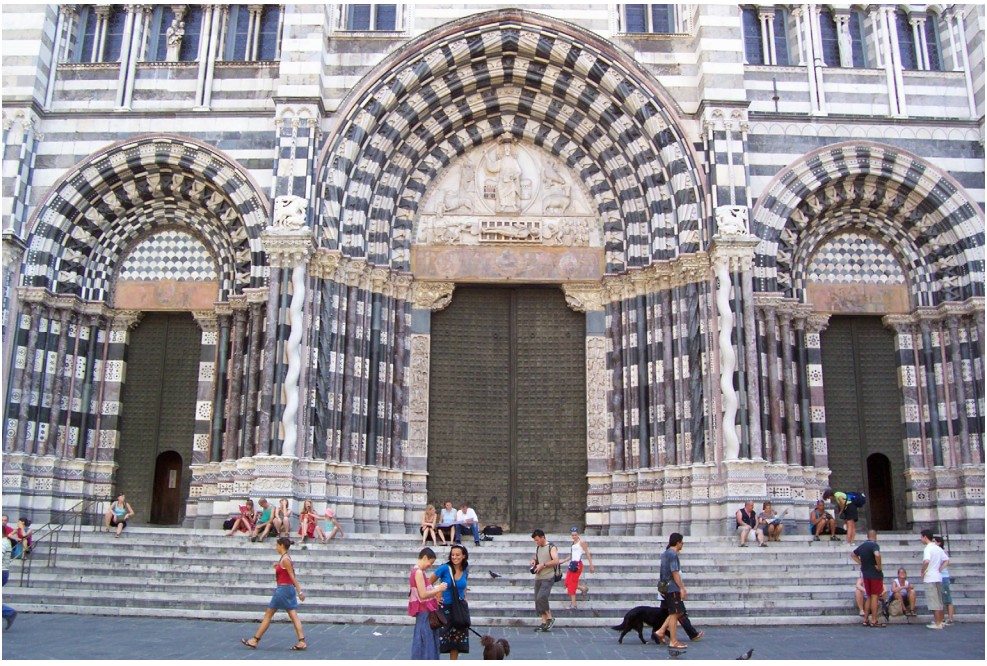

**Figure 2.** Façade of the cathedral in Genoa. It is composed of white marble, red limestone and green blocks of serpentinite. Picture courtesy of V. Sánchez Escribano.

When these rocks are quarried, dust from the extraction of the blocks can be dangerous for workers if the stones contain hazardous, fibrous components such as asbestos [8]; in fact, even the natural weathering of these rocks, if containing asbestos, can be a danger to the surrounding community [9,10]. Another closely related activity that can have hazardous

consequences is related to the field of restoration and heritage conservation. Laser is a major type of equipment used in such activities (e.g., cleaning), but the interaction with the stone can result in the liberation of stone particles that, if containing asbestos, could affect the professional workers when no safety equipment is used. The inhalation of fibres can provoke inflammation of the lungs and related illnesses, such as cancer, including lung cancer and mesothelioma.

For all these reasons, the detailed characterisation of serpentinites, focussing on their asbestos content, is of high interest. In this paper, we will focus on the emission of particles when using a laser in a stone (i.e., serpentinite) cleaning process.

This paper relates some scientific and social challenges that, at present, are included among the research priorities for the strategy and concerns to keep citizens around the world safe. This is part of Sustainable Development Goal 3 on Good Health and Well-being [11]. The presence of fibrous minerals in stone has a high economic impact, and stone companies often must cease extraction and exploration due to the potential hazardous effects of the stones. This is due to strict laws related to the asbestos content of rocks for the safety of workers [12]. In this paper, the authors present two samples containing fibrous chrysotile (asbestos), among other minerals. The main goal of the research was to apply laser ablation on fully characterised samples, implementing a filter in the laser equipment to determine how the different particles interact with the filter. Our results support the continuation of research on the further implications on health when preventive measures are not applied when working with fibrous materials.

## 2. Materials and Methods

Two slabs of commercial serpentinite were provided by a Spanish stone company (M-12 and M-17, Figure 3).

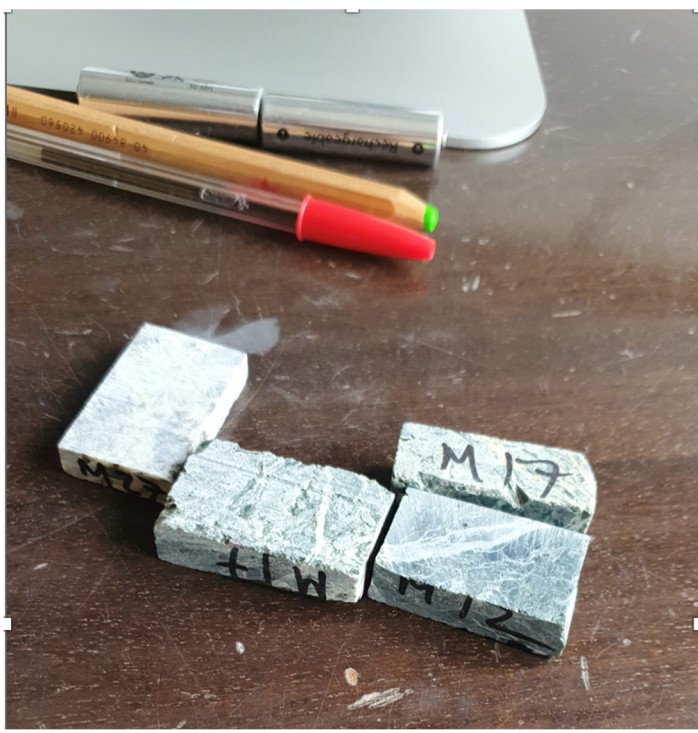

**Figure 3.** Slabs of commercial serpentinite used in our experiments.

To characterise the stones, we used inductively coupled plasma mass spectrometry (ICP-MS), laser ablation, petrographic microscopes, scanning electron microscopy with energy-dispersive spectrometry (SEM/EDS) and X-ray powder diffractometry (XRPD). After laser ablation, and in order to characterise the obtained powder, we used XRPD, differential scanning calorimetry and thermogravimetric analysis (DSC/TG) and transmis-

sion electron microscopy combined with energy-dispersive spectrometry and selected-area electron diffraction (TEM/EDS/SAED).

1. The chemical composition (whole rock major and trace elements) was obtained by means of inductively coupled plasma mass spectrometry (ICP-MS) using an ICP-MS AGILENT 7800. For the analysis, 0.1 g of powder from each sample was digested with $HNO_3$ + HF under pressure, in high-pressure vessels, in a Milestone microwave.

2. A complete petrographic examination was carried out, following standard ASTM C1721-15 [13], to describe the mineralogy and textures. A Leica DM2500P microscope under transmitted light was used for this purpose. The same microscope was used for the first examination of the powder after laser ablation.

3. X-ray powder diffractometry (XRPD) was used for the characterisation of the samples before (whole rock) and after (filtered particles) laser ablation. XRPD was performed using a Bruker D8 Advance (Bruker, Billerica, MA, USA) X-ray diffractometer at 40 kV and 40 mA. The instrument was equipped with a copper tube and a curved graphite monochromator. Scans were recorded in the range of 3–66 °2θ, with a step interval of 0.02 °2θ and a step-counting time of 3 s/step. EVA software (DIFFRACplus EVA) was used to identify the mineral phases and experimental peaks, being compared with the PDF2 reference patterns.

4. Differential scanning calorimetry (DSC) and thermogravimetric analysis (TG) were performed in an alumina crucible under a constant aseptic air flow of 30 mL min$^{-1}$ with a Netzsch STA 449 C Jupiter (Netzsch-Gerätebau GmbH, Selb, Germany) in a 25–1200 °C temperature range, with a heating rate of 10 °C min$^{-1}$. Approximately 30 mg of sample was used for each run. Instrumental precision was checked by five repeated collections on a kaolinite reference sample, revealing good reproducibility (instrumental theoretical T precision of ±2 °C), DSC detection limit <1 μW. Derivative thermogravimetry (DTG), derivative differential scanning calorimetry (DDSC), onset and exo- and endo-thermic peaks were obtained using the Netzsch Proteus thermal analysis software (Netzsch-Gerätebau GmbH, Selb, Germany). It is worth mentioning that the curve of differential scanning calorimetry (DSC) characterises the heat effects related to the physical and chemical conversion of samples. The thermogravimetric (TG) curve plots the weight variation of the sample during heating; the differential–thermogravimetric (DTG) curve characterises the rate of weight variation in the sample during heating.

5. The morphology of the samples after laser ablation was investigated with a transmission electron microscope (TEM) using a Jeol JEM 1400 Plus (120 kV) equipped with a Jeol large-area silicon drift detector SDD-EDS (Jeol, Tokyo, Japan) for microanalyses. Unambiguous crystallinity of individual fibres was achieved by selected-area electron diffraction (SAED). For TEM investigations, the sample was deposited on a Formvar carbon-coated copper grid.

6. The morphology of the samples before and after laser ablation was investigated via scanning electron microscopy (SEM) using an environmental scanning electron microscope, the FEI QUANTA 200 (FEI, Hillsboro, OR, USA), equipped with an X-ray EDS suite comprising a Si/Li crystal detector model, EDAX-GENESIS 4000 (EDAX, Tokyo, Japan).

7. Laser ablation: The laser used was a femtosecond pulsed near-infrared laser, specifically the Spirit system by Spectra-Physics, with emission wavelength 1040 nm and pulse width < 400 fs. The intensity profile at the laser output was near-Gaussian (M2 < 1.2) and the beam diameter at the exit of the laser head was 1.5 mm with horizontal polarisation (>100:1). The pulse rate can be set from a single shot to 1 MHz, with maximum pulse energy of 40 μJ at 100 kHz. The maximum mean power output is >4 W. A two-mirror galvanometric scanner (Raylase SuperscanIII-15) was used and scanned the laser beam in the X-Y directions. The beam was focused on the sample surface by means of a F-theta objective lens, 160 mm focal length, up to a diameter of roughly 30 μm.

To collect the dust generated during the laser ablation process, a plastic sample holder was produced by 3D printing (Figure 4). This sample holder device was manufactured ad hoc for this purpose, and it was coupled to the aspiration system of the laser laboratory. The mask that acted as a filter was located at the entrance of the suction tube; this was the reason that the dust was deposited directly on the outside face of the filter and did not pass through it, proving that this is a potential plausible protective measure for personnel who carry out laser cleaning.

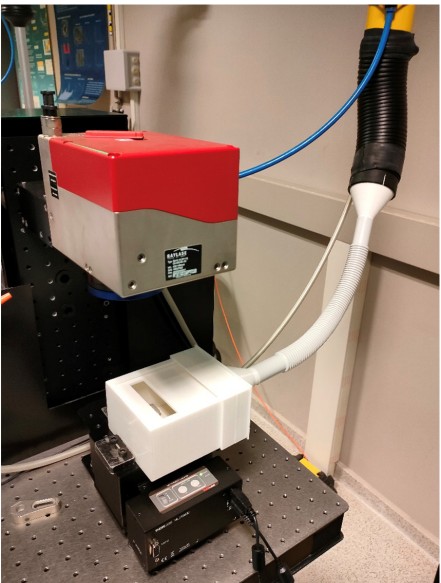

**Figure 4.** Ad hoc receptacle to collect sample during laser ablation.

When selecting the processing parameters, it was necessary to consider that our goal was to obtain, under laboratory conditions, sufficient powder to be analysed, and not to investigate the most suitable laser cleaning technique to remove unwanted crusts or layers, while preserving the integrity of the rock. Thus, the ablation process of the serpentinite samples was carried out in an area of 15 mm × 40 mm, with a pattern of horizontal and vertical lines, a separation of 20 mm and at a scan speed of 100 mm/s at the maximum laser power (100 W). A total of 20 repetitions of the ablation pattern were performed on the sample; therefore, the duration of the ablation process was approximately 2 h (Figure 5).

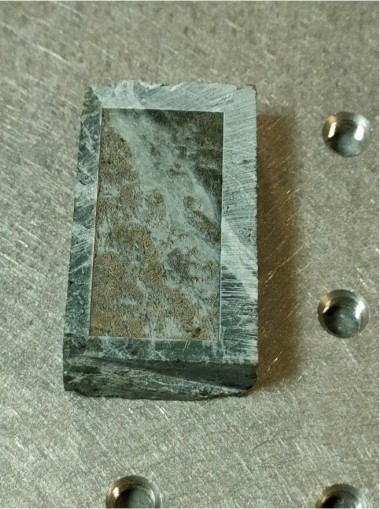

**Figure 5.** Rock sample after laser ablation.

## 3. Results

### 3.1. Petrography and ICP-MS

From the mineralogy and geochemistry of both samples, the authors could describe the slabs as composed of serpentinite with a major transformation to carbonates (Figure 6, Table 1). Carbonation is a very common characteristic of ornamental serpentinites, as they are easier to carve.

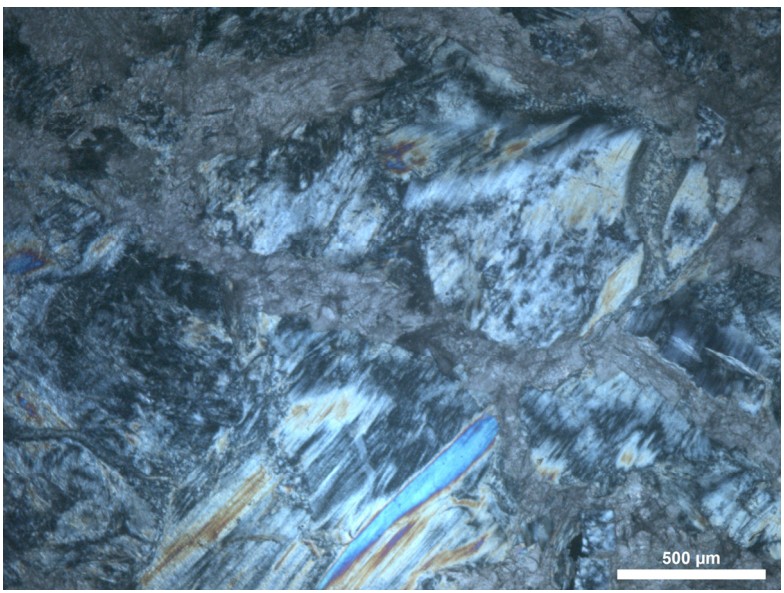

**Figure 6.** Sample M-12, produced from serpentine (both grey and blue-orange fibres) and carbonate (pinkish material covering the interstices of serpentine minerals). A Leica DM2500P microscope under transmitted light was used for this purpose at the University of Salamanca.

**Table 1.** Chemical analysis of samples M-12 and M-17. Major elements in %; trace elements in ppm.

| Ref. | Al$_2$O$_3$ | CaO | Fe$_2$O$_3$ | K$_2$O | MgO | MnO | Na$_2$O | P$_2$O$_5$ | SiO$_2$ | TiO$_2$ | LOI |
|------|------|------|------|------|------|------|------|------|------|------|------|
| M-12 | 5.65 | 3.08 | 9.15 | 0.12 | 30.29 | 0.18 | 0.52 | 0.16 | 34.57 | 0.41 | 14.17 |
| M-17 | 0.98 | 0.17 | 6.69 | 0.07 | 29.37 | 0.04 | 0.38 | 0.01 | 47.58 | b.d.l. | 15.80 |

| Ref. | Li | V | Cr | Co | Ni | Cu | Zn | Sr | Y | Ba | Pb |
|------|------|------|------|------|------|------|------|------|------|------|------|
| M-12 | 3 | 101 | 1461 | 63 | 849 | 2 | 38 | 40 | 3 | 2 | 1 |
| M-17 | 3 | 35 | 2409 | 83 | 1775 | 5 | 40 | 2 | 0.5 | 3 | 2 |

b.d.l. below detection limit. LOI: loss on ignition.

### 3.2. XRPD Characterisation

From a mineralogical point of view, sample M-12 was composed of serpentine, clinochlore, talc and dolomite as major phases (Figure 7), while sample M-17 was composed of talc, serpentine, magnesite and dolomite. Peaks for this sample were less evident, as explained below (Supplementary Materials S1) (Table 2).

After laser ablation, we obtained a powder that was retained in the filters (both High-Efficiency Particulate Air (HEPA) filter and hygienic face mask) (Figure 8). This powder was first checked under the microscope, and then was transferred to vials to be analysed by XRPD and SEM, DSC/TG and TEM/EDS/SAED to detect the possible presence of asbestos minerals such as chrysotile.

Raw sample M-17, when characterised by XRPD, showed all sharp peaks, indicating a high level of crystallinity of dolomite, magnetite talc and serpentine. After laser ablation, serpentine and magnesite were still detected by XRPD (Figure 9). In particular, serpentine showed reflections at (002), (004) and (−111), which were still visible due to texture effects, even with remarkably reduced intensity (Supplementary Materials S2). Concerning the M-

12 sample, after laser ablation, the reflections obtained by XRPD and attributed to dolomite were still visible with high intensity.

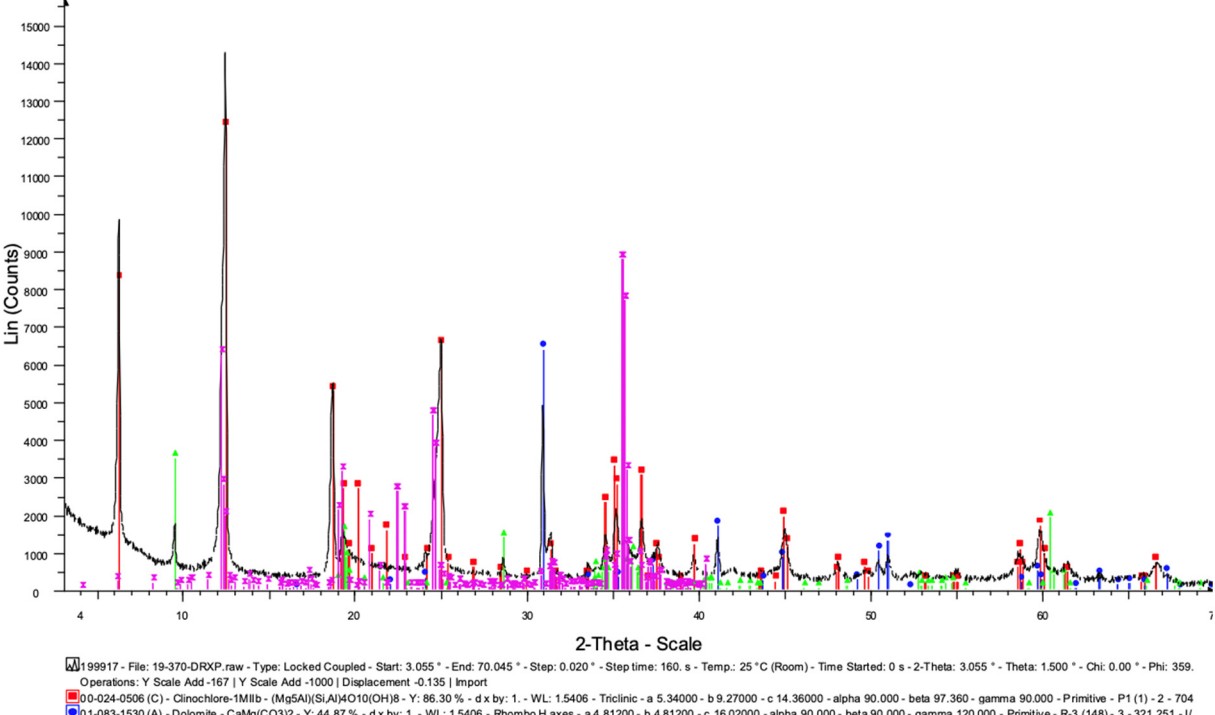

**Figure 7.** XRPD of sample M-12, whole rock before ablation, where peaks of a serpentine phase (antigorite) and carbonate (dolomite) are preeminent, also containing some talc and chlorite.

**Table 2.** Mineralogy of samples before (raw sample) and after laser ablation.

| Sample | Raw before Laser Ablation as Detected by XRPD and SEM/EDS | After Laser Ablation Phases as Detected by XRPD, DSC/TG, TEM/EDS/SAED |
|---|---|---|
| M-12 | Serpentine > clinochlore > talc > dolomite | Dolomite > talc > chrysotile> lizardite |
| M-17 | Talc > serpentine > magnesite > dolomite | Magnesite > talc > chrysotile |

### 3.3. DSC/TG Characterisation

To avoid ambiguity in the distinction of serpentine polymorphs, the M-17 and M-12 samples were also characterised by DSC/TG and TEM/EDS/SAED. It is worth mentioning that thermal analysis has been successfully employed in the distinction of serpentine minerals (chrysotile, lizardite, antigorite and polygonal serpentine) [14–16] that are otherwise subject to ambiguous characterisation by XRPD.

The thermograms of the two samples were different (Figure 10a–d) due to the different phases present, besides chrysotile. The DSC curve, relative to the sample M-17, showed a shoulder effect at around 610 °C, which became a very evident peak at 614 °C in the DDSC curve (Figure 10a), which was related to chrysotile collapse [15]. This was also confirmed by the maximum loss rate peaks recorded on the DTG curve at 611 °C, ascribed to chrysotile dehydroxylation (Figure 10b). It is worth mentioning that the dehydroxylation temperature was taken to be the temperature of the maximum rate of weight loss. The very intense dehydroxylation effect (between 500 and 600 °C) that preceded the collapse of the chrysotile (Figure 10b) was related to the breakdown of the magnesite [14].

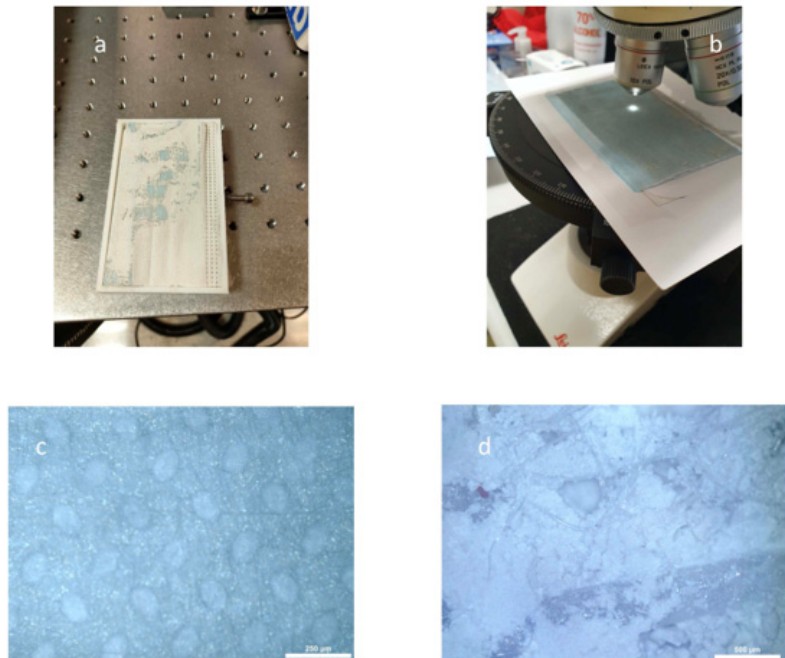

**Figure 8.** Study under the microscope of powder obtained after laser ablation. (**a**) Mask retrieved from the laser equipment; (**b**) mask under the microscope; (**c**) clean mask under the microscope—the original texture of the mask is shown here; (**d**) mask with the retained powder after laser ablation—only a mass of talc is seen in this picture.

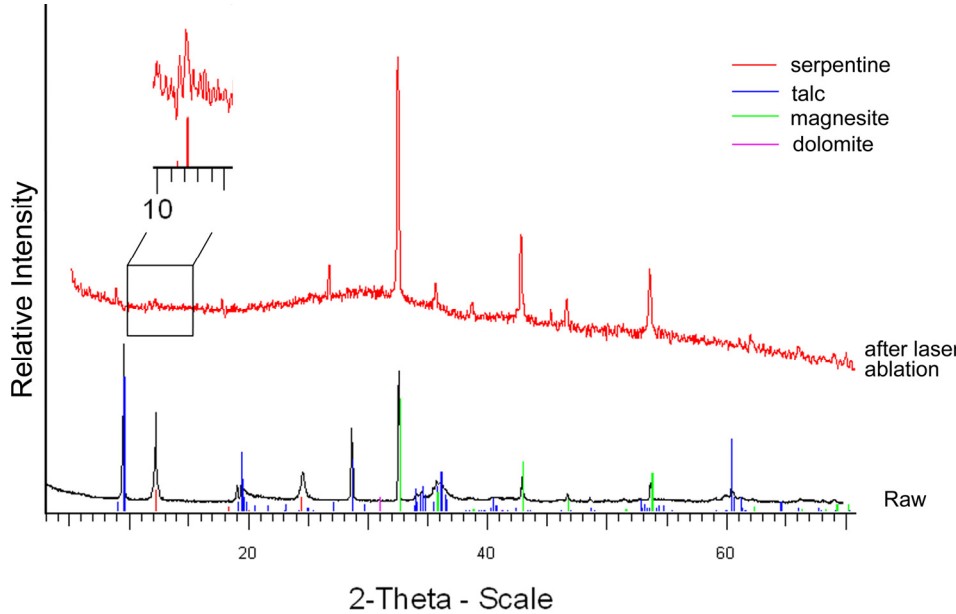

**Figure 9.** Diffractogram of sample M-17 before (raw) and after laser ablation; the black square in the figure indicates the enlarged diffractogram area underneath, where the main reflection of the serpentine mineral phase is clearly visible.

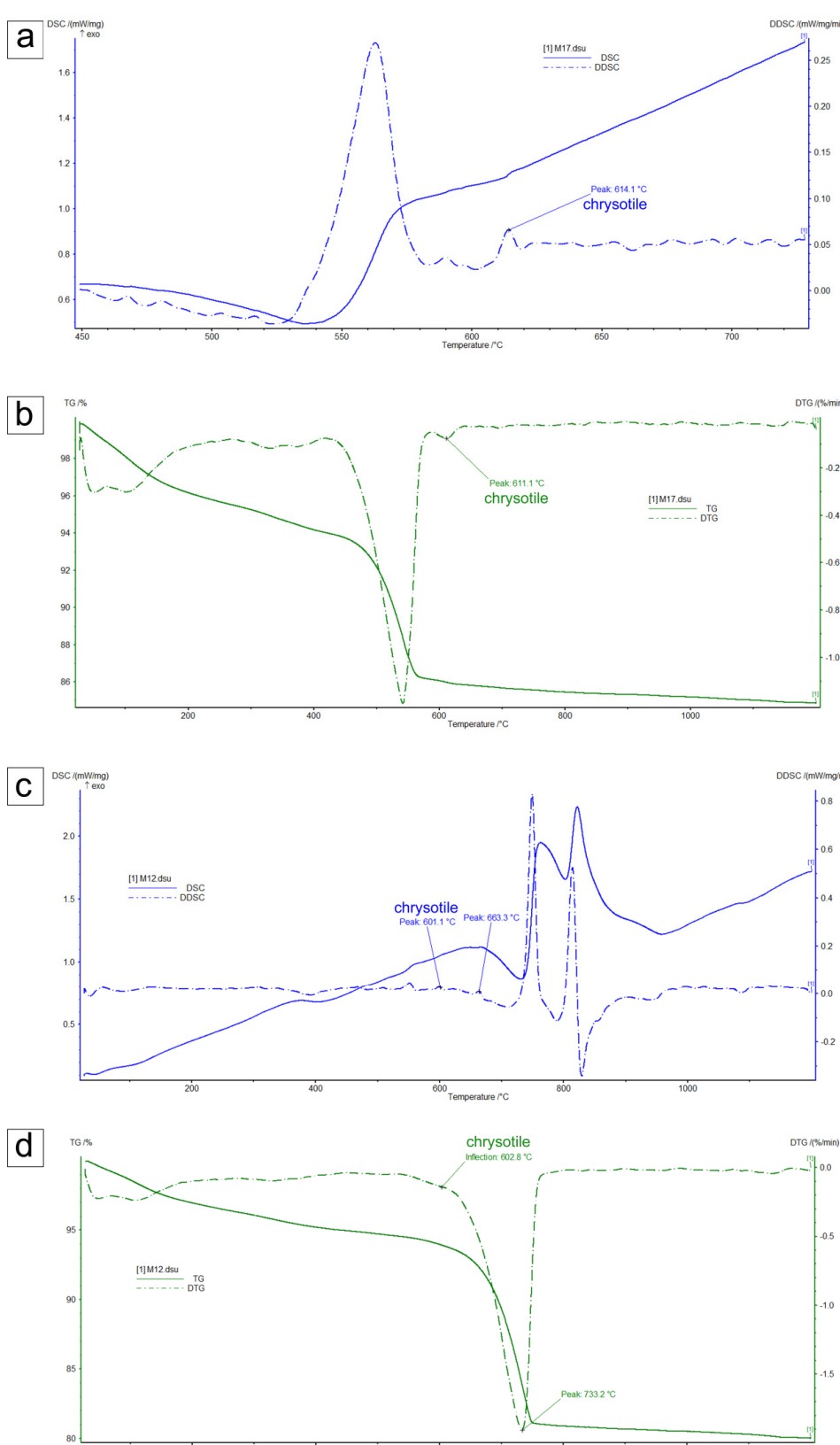

**Figure 10.** Thermal analysis of M-17 and M-12 samples: (**a**) sample M-17, blue solid line DSC, blue dashed line DDSC; (**b**) sample M-17, green solid line TG, green dotted line DTG; (**c**) sample M-12, blue solid line DSC, blue dashed line DDSC; (**d**) sample M-12, green solid line TG, green dotted line DTG.

DTG on sample M-12 showed one very intense maximum loss rate at 733 °C, ascribed to the liberation of $CO_2$ relative to the dolomite collapse, and a weak shoulder at 603 °C, which was ascribed to chrysotile breakdown (see Figure 10d). The weak signal at around 600 °C on the DDSC curve confirms the presence of chrysotile (serpentine polymorph) (Figure 10c). Moreover, at 663 °C, a signal was clearly recorded on the DDSC curve due to lizardite breakdown. In both thermograms, the low intensity of the peaks that identify chrysotile is related to the low quantity present in HEPA filters. Moreover, in both thermograms (Figure 10b,d), the initial weight loss observed up to 120 °C on the TG curves was attributed to the release of water adsorbed at the sample surface and on the DSC curve (Figure 10a,c), and the exothermic peaks were related to the crystallisation of phases (e.g., forsterite) after the breakdown events [15,16].

### 3.4. TEM/EDS/SAED Characterisation

In both samples (M-17 and M-12), at lower magnification, the TEM images showed mainly the residues of the filter, corpuscles of spherical morphology (Figure 11a), with also sporadic residues of the pre-existing phases, such as chrysotile, dolomite, magnesite and talc (Figure 11b). We mainly focused on the identification and characterisation of the asbestos phases, i.e., chrysotile. The longest fibres reached dimensions of more than 2 μm in length, with outer and inner diameters being, on average, 50 and 10 nm, respectively. This size is beyond the range of respirable fibres, defined by WHO [17] as fibres having width ≤ 3 μm, length ≥ 5 μm and length/width ratio ≥ 3:1. However, the aerodynamics of this motion are caused mostly by the fibres' width [18]. A more accurate size evaluation would require thousands of measurements under SEM and TEM. Unfortunately, chrysotile fibres are often curled up in polypropylene filter residues, and the fibres' sizes may be underestimated.

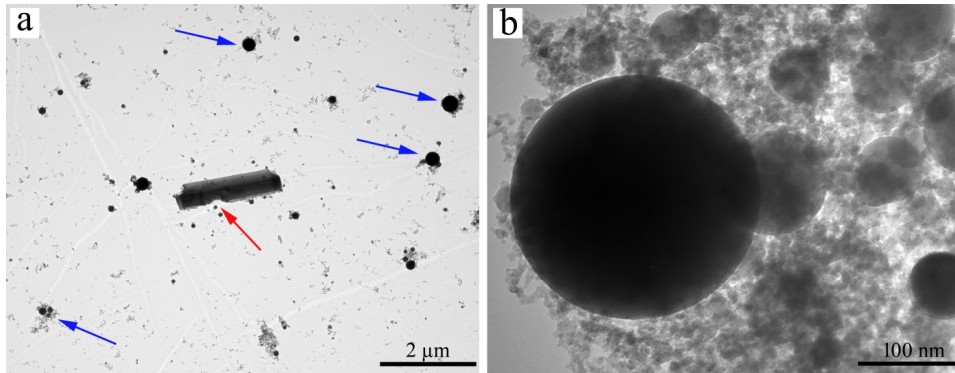

**Figure 11.** (**a**) Transmission electron micrographs of polypropylene filter residues indicated by blue arrow and talc indicated by red arrow; (**b**) spherical morphology of polypropylene filter residues.

At higher magnification, bundles of chrysotile could be observed, in which the individual fibres showed the classic tube shape (Figure 12a,c). All the observed chrysotile fibres showed a well-defined crystallinity, as revealed by the electron diffraction patterns of selected areas (SAED) (Figure 12b). The chemical data of the individual single fibres were consistent with the chemical composition of chrysotile ($Mg_3Si_2O_5(OH)_4$) (Figure 12b). Other minor and trace elements are known to be present in the chrysotile samples [16], as also confirmed by the geochemical data of the origin rock (Table 2). However, due to the instrumental limits, these elements were below the detection limit. Serpentine lizardite was crystallised in lamellar sheet form (plate-like morphology) and, as seen by TEM, showed easy splitting in the sheets (Figure 12d).

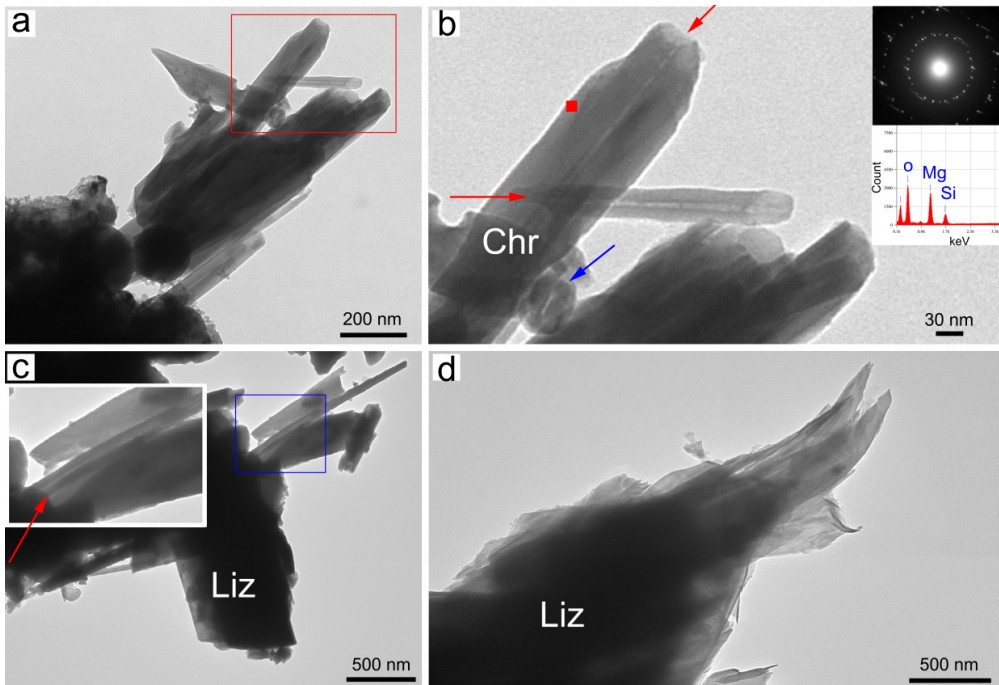

**Figure 12.** Transmission electron micrographs of samples M-17 (**a**,**b**) and M-12 (**c**,**d**). (**a**) The classic tube morphology of chrysotile can be observed. (**b**) Magnification of (**a**) indicated in the empty red square. The red arrows indicate the chrysotile core, which always runs empty longitudinally along the fibre's axis. The blue arrow indicates a chrysotile fibre section perpendicular to its direction of elongation. Top right chrysotile SAED and chemical analysis (EDS/TEM) both acquired at the point indicated by the solid red square. (**c**) Chrysotile and lizardite magnification indicated in the empty blue square. The red arrows indicate the chrysotile core. (**d**) Lizardite showing plate-like morphology.

## 4. Discussion and Conclusions

Among the minerals that form airborne particulates, the most hazardous ones display a fibrous–asbestiform crystal habit [18], such as chrysotile. The presence and the structural coordination of surface Fe are considered the main factors of fibre toxicity, together with bio-persistence and the aspect ratio [19–23]. Toxic effects on human health, due to exposure to high doses (essentially of the occupational type) to asbestos, have been recognised for a very long time [24].

However, the toxic effect due to environmental exposure at low doses of asbestos has also received a great deal of attention [25,26]. In recent research, attention has turned to the potential deleterious effects of naturally occurring asbestos (NOA) exposure. The airborne presence of asbestos fibres has been ascribed to their release from anthropic sources (e.g., from the structural and ornamental stones used in buildings), as well as from natural sources (e.g., from outcropping rocks), caused by a combination of atmospheric agents and anthropogenic activities. Pulmonary pathologies, verified in many different parts of the world, have been attributed to the rocks normally used in local buildings—for example, chrysotile and tremolite asbestos in outcropping ophiolitic rocks and soils in several regions of Italy and Spain [27–30]. The quantities of chrysotile detected in both samples after laser ablation are not negligible. The global scientific community has also acknowledged the lack of evidence of a threshold level of exposure to asbestos fibres, below which there is no risk of mesothelioma [28]. Rock scientists (petrographers and mineralogists) can provide detailed characterisations of asbestos minerals and determine the type of work environment exposure by analysing the powder left on HEPA filters and hygienic facemasks. Another aspect to consider is that the presence of trace elements, especially heavy metals, in asbestos minerals' structures contributes to their toxic potential [29] due to the capability of these elements to induce lung cancer, if released into the body once the asbestos is inhaled.

The authors of this paper believe that a continuation of the research is needed, using samples containing a much greater number of fibrous minerals, but, at present, there are some conclusions that can be considered, as chrysotile (asbestos) was detected in both powder samples coming from the ablation laser of M-12 and M-17:

- Asbestos exhaled during laser ablation processes (e.g., cleaning, texturizing, etc.) could represent a serious hazard for the environment and for occupational safety.
- HEPA filters and hygienic face masks retain particles of different nature, expelled through laser cleaning. It is easier to investigate this using hygienic face masks, due to their texture.
- Either the use of masks by workers or the implementation of a filtering method in the ablation equipment can reduce the risk of fibre inhalation during cleaning activities to protect and restore monuments and historical buildings.
- Communicating these results to workers during outreach activities will encourage self-protection measures, which will protect their health from these hazardous materials. A precise explanation of how these fibres can be lethal if no prevention is implemented can save lives.

We conclude that using the correct information and instructing workers on the use of appropriate face masks as a mandatory part of their occupational safety protocol could help to reduce the risk of inhalation of asbestos fibres, which are highly harmful to health. In order to obtain more data on the release of chrysotile and/or other asbestos minerals from the laser ablation of serpentinites, more experiments are underway to facilitate advances in health and safety at work.

**Supplementary Materials:** The following supporting information can be downloaded at: https://www.mdpi.com/article/10.3390/app13010043/s1, S1 XRPD of raw M17 sample before laser ablation; S2 XRPD of raw M17 sample before and after laser ablation; green arrow indicates main serpentine reflection.

**Author Contributions:** Conceptualization, D.P. and A.B.; methodology, D.P., A.J.L., A.R. and A.B.; formal analysis, D.P. and A.B.; investigation, D.P., A.J.L., A.R. and A.B.; resources, D.P., A.J.L., A.R. and A.B.; writing—original draft preparation, D.P. and A.B.; writing—review and editing, D.P., A.J.L., A.R. and A.B.; funding acquisition, D.P., A.J.L., A.R. and A.B. All authors have read and agreed to the published version of the manuscript.

**Funding:** This research was funded by the University of Salamanca, grant number 2016/00218/001; the Spanish Ministry of Science and Technology, grant number PID2021-123948OB-I00; and Ministero italiano dell'Università e della Ricerca (MIUR) Progetti di Ricerca di Interesse Nazionale (PRIN) Italy 20173 × 8WA4.

**Institutional Review Board Statement:** Not applicable.

**Informed Consent Statement:** Not applicable.

**Data Availability Statement:** Not applicable.

**Conflicts of Interest:** The authors declare no conflict of interest.

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
