# Peer review of "The Importance of Prevention When Working with Hazardous Materials in the Case of Serpentinite and Asbestos When Cleaning Monuments for Restoration"

_applsci, doi:10.3390/app13010043_

Round 1

Reviewer 1 Report

The article focuses on the important topic of health protection while handling a particular type of rock. It mentions the danger it poses while quarrying it but also when cleaning monuments made of this stone as a part of conservation treatment.

It is generally well written but some alterations are proposed:

The introduction lacks a proper mineralogical description of the rock itself. Please provide one.

Figures 1 and 2: Are both images (of serpentinite containing monuments? Please clarify.

Line 50: Use reference system adopted by the journal. 

Line 61: Improve English on this sentence.

Line 71: ?? Where?? When referencing use numbers in brackets

Line 73: Conclusions should not be stated at this point.

Figure 6: Where and under which conditions was this picture taken? Specify the model of the equipment, either here or in the materials and methods section.

Figure 7: Peaks

Figure 7: And what about sample M-12? Show the graph as well.

It is not clear how the dust settled on the face mask, since it was such a directed beam. Please explain. And was there dust on both sides of the mask? Can this mask offer protection to this particular problem? Elaborate on the conclusions.

Figure 8: What do we see in images c and d? Provide a description.

Figure 10: Add the sample name in the b) and d) captions.

Line 83: Remove e.g.
Provide as many references as needed to support this fact.
Check the remaining text.

References: Too many self citations (8/21).
Improve and expand bibliography.

Author Response

Dear reviewer:

Thank you very much for the detailed review. We have gone through all your questions, suggestions and corrections and were able to implement all of them. Following there is a description of how we did that.

-The introduction lacks a proper mineralogical description of the rock itself. Please provide one.

We have tackled this issue by including some description as follows: From the mineralogical point of view, serpentine-group minerals (i.e. chrysotile, lizardite, antigorite) are the main constituents of serpentinite rocks (Deer et al., 2009). Although serpentinite rocks often may also contain accessory minerals such as chlorite, brucite, magnetite, talc or carbonates as well as scraps of the predecessor mineralogy such as pyroxene and olivine (Deer et al., 2009). Serpentinites have a complex origin, involving deep parts of the Earth, including the mantle. They are formed as a result of  serpentinization of ultramafic rocks (i.e. lherzolite, harzburgite and dunite) in which the main minerals are pyroxenes and olivine. Antigorite, lizardite and chrysotile are polymorphs with a similar chemical composition, close to the magnesian end-member Mg3Si2O5(OH)4 (Deer et al., 2009).

- Figures 1 and 2: Are both images (of serpentinite containing monuments? Please clarify.

The main material used to build this historic buildings was serpentinite. The legend of the figures has now an explanation.

- Line 50: Use reference system adopted by the journal. 

done

- Line 61: Improve English on this sentence.

The line has been reworded.

  •  
  • Line 71: ?? Where?? When referencing use numbers in brackets

Reference has been added.

- Line 73: Conclusions should not be stated at this point.

The referred statement has been moved to the Discussion and Conclusions section.

- Figure 6: Where and under which conditions was this picture taken? Specify the model of the equipment, either here or in the materials and methods section.

Although the information of the microscope was already in the Methodology section, we have included the information in the legend of this figure also.

- Figure 7: Peaks

Corrected

- Figure 7: And what about sample M-12? Show the graph as well.

The problem with M-12 is that the content of serpentine minerals is too low to show in a diffractogram. For this reason, we though it was better to include it in the supplementary materials. Explanation on this is also included in the text.

- It is not clear how the dust settled on the face mask, since it was such a directed beam. Please explain. And was there dust on both sides of the mask? Can this mask offer protection to this particular problem? Elaborate on the conclusions.

A better explanation of how the powder is stoped by masks has been included. The powder does not filter to the other side of the mask. For this, we consider this is a potential heath protection and we have elaborate this in the text, highlighting the advance in occupational safety.

- Figure 8: What do we see in images c and d? Provide a description.

An explanation has been included in the legend of the figure.

- Figure 10: Add the sample name in the b) and d) captions.

done

- Line 83: Remove e.g.

done

- Provide as many references as needed to support this fact.

done

Check the remaining text.

checked

References: Too many self citations (8/21).
Improve and expand bibliography.

Some self-citations have been deleted and other references have been added for a complete bibliography on all the tackled issues.

Reviewer 2 Report

Dear Authors,

Thank-you for a very interesting article which I enjoyed reading. It is really important that you have highlighted a potential risk in this industry.

My main concern with the article lies around the advice to workers to use facemasks to reduce risks. Whilst I agree that there could be a significant risk and that your analysis clearly demonstrates the liberation of asbestos fibres, best practise asbestos management would require more than just a simple facemask to be used.  I am not at all convinced that the paper provides enough evidence to support the conclusion that "filters are an effective and cheap prevention measure ....... even eliminate, the risk of fibre inhalation" and I strongly recommend that this is removed from the article. Asbestos management should involve specific ppe (inc. masks), and decontaminaton procedures - which you could refer to if you wanted to provide advice but simply wearing a non-specified mask is not enough. The same is true for sentences 73-74 in the introduction (because there is insufficient proof that this action could save lives and in fact, the advice could be dangerous). I have scored scientific soundness as low, not because the analysis but because of the links to health outcomes.  I think if the paper is simply about the risks and the liberation of chrysotile with strong recommendations to research further - this would be great and the score improve greatly.

I have a few suggestions for areas of improvement as explained below, section by section

Introduction

This reads fine but I would recommend altering the focus to concentrate less on the health risks and more on the mineralogy - for example Lizardite is not mentioned until p8 and the general reader may not be familiar with this mineral? It maybe useful for the reader to know that lizardite and chrysotile have the same chemical composition.

Sentence 32 - "Serpentinites are important rocks...." could do with a reference

Sentences 36-37 - I find this sentence a little vague. What do you mean by specific nature? I can see that Serpentinites are widely used and have been identified by the IARC as hazardous but what isn't known?

Sentence 50 - do you have a reference for this statement "actually even the weathering of these rocks, if containing asbestos, can ......."

Sentence 55-56 - Mesothelioma is a type of cancer so suggest change this to something like "....related illnesses, such as cancer, including lung cancer and mesothelioma"

Sentences 58-59 - are  very similar to previous content (and a little vague)

Sentences 62- is there any evidence of these "very dramatic health issues" from monument restoration ? ref ? 

Sentence 68 - please reference the UN SDG's

Sentence 68 - why does the presence of the fibrous minerals in the stone affect economic impact? I think that I understand that you are suggesting that by not using this stone (because of health impacts) - this will have a high impact on the economy?  Is this true ? what is the economic value of this industry?

Sentences 73-74 - please remove as this research cannot support this statement

Methods

Very easy to follow but I do think that the table formatting could be improved and I am a little confused by Table 2. Please can you write a statement to explain why the major phases present in the samples changed after ablation? also can you explain "chrysotile = lizardite" ? (are you referring to chemical comp?)

Please can you also rephrase the heading for Figure 9 as it is not clear.

Discussion and conclusions

Sentences 89-90 - what do you mean by "last type of particles"?

Sentences 101-102 - I do not think that this is an outcome of this research which has not measured health outcomes.

Sentences 112-113 - suggest "Asbestos exhaled during laser ablation processes could represent a serious danger for the environment......."

General

Do you have any comments about the characteristics of the fibres which are liberated during cleaning etc.? are they of a respirable size?

Author Response

Dear reviewer:

Thank you very much for your suggestions and corrections to improve this paper. We have gone through  all of them and we have been able to tackle them in the following way:

- My main concern with the article lies around the advice to workers to use facemasks to reduce risks. Whilst I agree that there could be a significant risk and that your analysis clearly demonstrates the liberation of asbestos fibres, best practise asbestos management would require more than just a simple facemask to be used.  I am not at all convinced that the paper provides enough evidence to support the conclusion that "filters are an effective and cheap prevention measure ....... even eliminate, the risk of fibre inhalation" and I strongly recommend that this is removed from the article. Asbestos management should involve specific ppe (inc. masks), and decontaminaton procedures - which you could refer to if you wanted to provide advice but simply wearing a non-specified mask is not enough. The same is true for sentences 73-74 in the introduction (because there is insufficient proof that this action could save lives and in fact, the advice could be dangerous). I have scored scientific soundness as low, not because the analysis but because of the links to health outcomes.  I think if the paper is simply about the risks and the liberation of chrysotile with strong recommendations to research further - this would be great and the score improve greatly.

You are right, we overestimated  the protection for the workers based only on wearing a mask. Wearing a mask, from our results, will highly increase the protection, but it is not clearly the solution for an important occupational problem. So we have followed your advise and change the wording along the manuscript, including the  conclusions, advising the use of face masks as part of a more strict safety protocol.

I have a few suggestions for areas of improvement as explained below, section by section

Introduction

This reads fine but I would recommend altering the focus to concentrate less on the health risks and more on the mineralogy - for example Lizardite is not mentioned until p8 and the general reader may not be familiar with this mineral? It maybe useful for the reader to know that lizardite and chrysotile have the same chemical composition.

This has been done and a explanatory sentence on mineralogy has been added in the Introduction section.

- Sentence 32 - "Serpentinites are important rocks...." could do with a reference

Important references have been added.

- Sentences 36-37 - I find this sentence a little vague. What do you mean by specific nature? I can see that Serpentinites are widely used and have been identified by the IARC as hazardous but what isn't known?

Vague expression has been removed. Now the term asbestos has been explained and included in the text, as this is really the hazardous part of the rock.

- Sentence 50 - do you have a reference for this statement "actually even the weathering of these rocks, if containing asbestos, can ......."

We have been added a couple of sentences to illustrate this  issue.

- Sentence 55-56 - Mesothelioma is a type of cancer so suggest change this to something like "....related illnesses, such as cancer, including lung cancer and mesothelioma"

We have implemented your suggestion.

- Sentences 58-59 - are  very similar to previous content (and a little vague)

We have modified the text to avoid vague expressions and duplications.

- Sentences 62- is there any evidence of these "very dramatic health issues" from monument restoration ? ref ? 

We have reworded this part. We do not know of any "very dramatic health issue in monument restoration", although big precaution should be taken.

- Sentence 68 - please reference the UN SDG's

Included.

- Sentence 68 - why does the presence of the fibrous minerals in the stone affect economic impact? I think that I understand that you are suggesting that by not using this stone (because of health impacts) - this will have a high impact on the economy?  Is this true ? what is the economic value of this industry?

Serpentinites are very appreciated as ornamental stones because the distinctive colours and textures, but in some places they are not extracted any more (e.g. Spain, because law restrictions related to fibrous minerals). Most serpentinites are imported from India, Pakistan, Guatemala... The extraction of serpentinite could involve a health hazard for the workers. Closing a quarry involves a big social and economic damage. 

- Sentences 73-74 - please remove as this research cannot support this statement

 Ok, the sentence has been removed

Methods

Very easy to follow but I do think that the table formatting could be improved and I am a little confused by Table 2. Please can you write a statement to explain why the major phases present in the samples changed after ablation? also can you explain "chrysotile = lizardite" ? (are you referring to chemical comp?)

Table 2 has been edited.

Yes, chrysotile = lizardite referred to the same chemical composition. However in order to clarify, we have changed the writing to: chrysotile > lizardite.

- Please can you also rephrase the heading for Figure 9 as it is not clear.

Rephrased.

Discussion and conclusions

- Sentences 89-90 - what do you mean by "last type of particles"?

 asbestos fibres. We have changed the wording.

- Sentences 101-102 - I do not think that this is an outcome of this research which has not measured health outcomes.

The speculative expression has been removed

- Sentences 112-113 - suggest "Asbestos exhaled during laser ablation processes could represent a serious danger for the environment......."

 done

General

- Do you have any comments about the characteristics of the fibres which are liberated during cleaning etc.? are they of a respirable size?

Unfortunately, we have not explored this aspect yet. It will need the analysis of hundreds of measures to have a reasonable statistical analysis. At the moment we are working with a larger set of samples and we are confident to have appropriate results soon to be able to answer your question. But we have added few lines to better describe this issue.

Round 2

Reviewer 2 Report

Thanks for making the changes - please can I ask for one final addition before publication

Discussion and Conclusions

line 365

"We conclude that using the correct information and instructing workers on the use of appropriate facemasks as a mandatory part of their occupational safety procedure"

because it is important to not rely on the use of simple facemasks when dealing with airborne asbestos.

Otherwise- all good to go - changes look great

Author Response

Dear reviewer,

Thank you very much for your suggestion. It has been implemented and the text looks much better now with your help.

Best wishes.